# Consequences of Thermal Variation during Development and Transport on Flight and Low-Temperature Performance in False Codling Moth (*Thaumatotibia leucotreta*): Fine-Tuning Protocols for Improved Field Performance in a Sterile Insect Programme

**DOI:** 10.3390/insects13040315

**Published:** 2022-03-23

**Authors:** Elizabeth J. Huisamen, Minette Karsten, John S. Terblanche

**Affiliations:** Department of Conservation Ecology and Entomology, Stellenbosch University, Stellenbosch 7602, South Africa; eopperman@sun.ac.za (E.J.H.); minettek@sun.ac.za (M.K.)

**Keywords:** fluctuating thermal regime, developmental acclimation, sterile insect technique, flight performance, false codling moth, cold tolerance

## Abstract

**Simple Summary:**

In two separate experiments, we examined how (1) developmental temperature and (2) transport conditions influenced the low-temperature performance and flight ability of false codling moth (FCM) adults. In experiment 1, larvae were exposed to a control (constant 25 °C), a cold treatment (constant 15 °C) or a fluctuating thermal regime (FTR) (25 °C for 12 h; 15 °C for 12 h) for 5 days, whereafter larvae were returned to 25 °C to pupate and emerge. After adult emergence, cold tolerance, life history traits and flight ability were scored. In experiment 2, adult FCM were exposed to 4 or 25 °C, with or without vibrations, whereupon flight ability, spontaneous behaviour (i.e., muscle coordination by monitoring whether the moth moved out of a defined circle or not) and cold tolerance were determined. The results of the first experiment showed that FTR led to enhanced cold tolerance, increased flight performance and high egg-laying capacity with minimal costs. The results of the second experiment showed that transport conditions currently in use did not appear to adversely affect flight and cold performance of FCM. These results are significant for refining conditions prior to and during release for maximum field efficacy in an SIT programme.

**Abstract:**

Here we aimed to assess whether variation in (1) developmental temperature and (2) transport conditions influenced the low-temperature performance and flight ability of false codling moth (FCM) adults in an SIT programme. To achieve the first aim, larvae were exposed to either a (control) (constant 25 °C), a cold treatment (constant 15 °C) or a fluctuating thermal regime (FTR) (25 °C for 12 h to 15 °C for 12 h) for 5 days, whereafter larvae were returned to 25 °C to pupate and emerge. After adult emergence, critical thermal minimum, chill coma recovery time, life history traits and laboratory flight ability were scored. For the second aim, adult FCM were exposed to 4 or 25 °C with or without vibrations to simulate road transportation. After the pre-treatments, flight ability, spontaneous behaviour (i.e., muscle coordination by monitoring whether the moth moved out of a defined circle or not) and chill coma recovery time were determined. The first experiment showed that FTR led to enhanced cold tolerance, increased flight performance and high egg-laying capacity with minimal costs. The second experiment showed that transport conditions currently in use did not appear to adversely affect flight and low-temperature performance of FCM. These results are important for refining conditions prior to and during release for maximum field efficacy in an SIT programme for FCM.

## 1. Introduction

The false codling moth (FCM *Thaumatotibia leucotreta* (Meyrick) (Lepidoptera: Tortricidae)) is a polyphagous lepidopteran pest species that infests several fruit crops such as citrus, apricots, pomegranates, prickly pears and avocados. It is native to sub-Saharan Africa and is present in South Africa, Zimbabwe, Angola, Gambia, Sudan, Israel and the islands of Madagascar, Réunion and Mauritius [1]. It is listed by the European and Mediterranean Plant Protection Organization (EPPO) as an A2 quarantine pest (pests that are present in EPPO countries, see a list of EPPO countries in [2]) and, thus, restrictions are placed on the international trade of its host crops [1]. Several control methods are employed including insecticide application, biological control agents (such as parasitoids, entomopathogenic nematodes (EPNs), entomopathogenic fungi (EPFs) and viruses), mating disruption and the sterile insect technique (SIT) [3]. However, insecticide resistance development has been observed in FCM (particularly to benzylurea compounds) and restrictions on the use of pesticides on exported citrus products [4,5] have highlighted the need for alternative control strategies. As a result, an SIT programme was initiated in South Africa in 2002 to manage FCM populations, initially in the Western Cape and later in the Eastern and Northern Cape provinces [6]. It involves mass rearing FCM, treating adults with ionizing radiation and releasing these (millions per week) into the wild population [5]. In FCM, ionizing radiation only partially sterilises the males but fully sterilises the females. The partially sterile males then mate with wild females in the released field, which subsequently lays infertile eggs (this is termed inherited sterility) [7,8]. The goal of the programme is to reduce the wild population of FCM through the use of SIT and to release partially sterile males at an overflooding ratio of at least ten sterile males to one wild male [7]. This maximises the probability that a wild female will mate with a sterile male in the field, leading to less fertile offspring and eventual population decline [3].

SIT programmes rely on the production of insects of high quality and performance to allow for the released males to outcompete the wild males [9,10]. The quality and performance of these insects are often tested by looking at several traits including thermal performance (critical thermal maximum (CT_MAX_) and critical thermal minimum (CT_MIN_), chill coma recovery (CCRT) and lower lethal limits (LLT)), life history (fecundity, developmental time and survival) and flight performance traits (at several temperatures) [10]. Dispersal and physiological performance of adult insects in the field can be influenced by current and past temperature conditions and as a result, changes in larval rearing temperature can induce changes in adult wing and body morphology and the ratios among these (e.g., wing-loading) [3,11,12]. Diamondback moth *Plutella xylostella* (Linnaeus) (Lepidoptera: Plutellidae) larvae reared at low temperatures develop into larger moths with longer forewings that have the ability to fly further in the field [13]. In *Drosophila melanogaster* (Meigen) (Diptera: Drosophilidae), cold-rearing larvae improved adult cold flight performance in the laboratory and field through changes in wing and body morphology and the ratios between these [12,14,15]. In the codling moth *Cydia pomonella* (Linnaeus) (Lepidoptera: Tortricidae), cold acclimation of larvae increased released adult field recapture rates at low temperatures but compromised high-temperature performance [16]. Additionally, codling moth adults exposed to four hours of transport had significantly decreased field recapture rates, indicating a possible impact of transport on flight ability [17]. Cold storage of adults before release (at 2 °C) also has a negative impact on their flight [18,19]. Given what is known from studies of other taxa, it is apparent that differences in larval rearing temperature and adult transport conditions in an SIT programme can influence adult flight ability in the field. Here, we wanted to better understand if changes in rearing temperature and transport conditions in specific life stages impacted the performance of adult FCM and if this might be a viable method to enhance adult field performance or at least offset some sub-optimal adult field recapture rates reported [20,21]. Particularly, since adult FCM lack rapid cold hardening [22] (a rapid improvement in survival at a lethal temperature after a short temperature pre-treatment (1–2 h) [23]) and, therefore, an increase in cold performance and an increase in flight performance at low temperatures induced by changes in rearing temperature will greatly increase the quality of released FCM.

The current FCM SIT protocol involves rearing larvae, pupae and adults at a constant temperature (25 °C). Upon adult emergence, they are irradiated at 200 Gy and then rapidly chilled and kept between 6 and 8 °C. This allows easier handling and transport prior to field release [20,24]. Unfortunately, this rapid chilling and prolonged low-temperature exposure leads to decreased field performance in laboratory reared FCM [25,26]. Furthermore, Boersma et al. [6] found that adult fecundity and longevity were affected by larval rearing temperature. However, the study by Boersma et al. [6] only employed constant rearing temperatures whilst insects in nature typically experience fluctuating temperatures. They also did not explore the possible impact of rearing temperature changes on adult flight performance. Exposing insects to a fluctuating thermal regime (FTR) during larval development within a non-detrimental temperature range could have a significant positive impact on several performance related traits (development time, fecundity, morphology and low-temperature performance) in adult insects [27,28]. Therefore, implementing FTR in an SIT mass rearing programme could provide the performance benefits of cold acclimation (increased performance in the field at cooler and fluctuating temperatures) without incurring a cost of prolonged exposure to cold temperatures (longer developmental times) [28,29].

Additionally, the field performance of adult mass-reared insects can be influenced by the conditions experienced by the adult insect during transportation to the release site. Several studies have shown decreased flight performance due to the effects of overcrowding, low-temperature storage, vibration (engine of the motorised vehicle) and time under hypoxia of the insects during transportation [20,30,31,32]. In the FCM SIT programme, the adult insects experience both vibration and low temperature conditions during transportation from the XSIT mass-rearing facility (X Sterile Insect Technique (Pty) Ltd. (XSIT)) in Citrusdal to the target areas for release, which could impact adult flight and low temperature performance in the field [20]. As mentioned previously, adult FCM lack rapid cold hardening (after exposure to several low temperatures (−6 °C to 0 °C) for several periods (2–10 h)) and, thus, the species is unable to adjust its thermal tolerance rapidly, which may reduce the performance of the sterile moths in the field after they were exposed to potential sub-optimal temperatures during transport [21,22]. Boersma et al. [6] found that adult cold storage prior to release decreased both dispersal ability and distance dispersed of released FCM. Nepgen et al. [24] showed a significant decrease in flight ability between adult non-transported, non-irradiated FCM and irradiated adult FCM transported for a long distance (12 h). However, these differences could be due to the presence of a combination of factors including radiation status of moths, vibration during transport, low-temperature exposure during transport, handling pressures, exposure to fluctuating temperature during packing and unpacking and the long duration of the trip [25,33]. Therefore, it is important to disentangle the different factors present during transportation that could have an effect on flight performance of FCM in the field and to determine whether the exposure to low temperature and/or the vibration experienced during transportation (alone or combined) have an effect on the flight performance of FCM. Additionally, Nepgen et al. [24] only looked at the effect of transportation on adult FCM after long distance transportation; however, short distance transportation could also potentially impact the flight ability of adults in the field and, therefore, needs to be investigated.

The aim of our study was to assess two distinct but important aspects of the mass-rearing and release production chain that may be used in an SIT programme: (1) how larval developmental temperature, and especially a fluctuating thermal regime, affects subsequent adult flight ability and low-temperature performance and (2) how adult transport conditions (vibration and low-temperature exposure during short periods of transport) influences flight and low-temperature performance in FCM. 

## 2. Materials and Methods

### 2.1. Developmental Plasticity and Performance

#### 2.1.1. Larval Acclimation Treatments on Adult Development and Performance

FCM larvae (fifth instar) were obtained from the XSIT mass-rearing facility and subsequently reared (on an artificial maize-based diet obtained from XSIT [34]) concurrently for five days at three different acclimation treatments (constant 15 °C, constant 25 °C and a fluctuating regime (fluctuating between 15 and 25 °C every 12 h)) (~1400 larvae per treatment) at 50% R.H. and a 12:12 h day/night cycle placed in three temperature-controlled incubators (one for each treatment)) (MRC LE-509, Holon, Israel). After the five-day acclimation period, the larvae were reared at 25 °C as per Stotter and Terblanche [22]. The fluctuating temperatures chosen (15–25 °C) are representative of the autumn/winter growing season in South Africa and cover the temperatures that FCM typically experience in the field [6,28,35] (Figure 1). Due to the slow emergence rates, all trials were run on 1–3 day old individuals to assure enough individuals were available per trial.

#### 2.1.2. Life History

To determine how larval developmental acclimation impacted the life history traits of adult FCM, daily egg-laying capacity and the percentage of adult emergence were estimated. Egg-laying capacity was determined by placing adult pairs of FCM in 30 replicate glass jars (500 g) per acclimation treatment, after which they were allowed to mate for 24 h (no food or water provided). The females were then allowed to lay eggs on baking paper for a further 24 h, after which the eggs were counted. Percentage adult emergence was determined by separating 50 larvae from each acclimation treatment (transferred with a soft bristled paint brush) to a container filled with sand (L: 50 cm, H: 38 cm, W: 37 cm) placed in an incubator at constant 25 °C, 50% R.H. and a 12:12 h day/night cycle (MRC LE-509, Holon, Israel). They were then allowed to pupate and the number of adult individuals that emerged were counted. 

#### 2.1.3. Low-Temperature Performance

The impact of developmental acclimation on the low-temperature performance of FCM was determined by testing critical thermal minimum (CT_MIN_) and chill coma recovery time (CCRT). 

Critical thermal minimum was estimated by placing individual moths (15 males and 15 females from each acclimation treatment) closed inside 1.5 mL microcentrifuge tubes on a Styrofoam “boat” (a platform made from Styrofoam on the sides (allowing it to float) and a clear plastic bag at the bottom, allowing temperature transference) (37 cm length × 29 cm width × 7 cm height) in a water bath (cw410-wl; Huber, Germany) containing ethanol. After an initial 15 min equilibration period at 25 °C, the temperature in the water bath was ramped down at 0.25 °C/min until the physiological coma-inducing endpoint of each moth was reached, which was indicated by a cessation of movement and no response to gentle prodding with a fishing line [21]. 

Chill coma recovery time was determined by placing individual moths in 1.5 mL microcentrifuge tubes, capping them and placing them in an ice slurry at 0 °C for two hours. The temperature in the slurry was confirmed with a thermocouple (36 standard wire gauge, Type T, Omega, Norwalk, CT, USA) connected to a handheld digital thermometer (Fluke 52 II, Eindhoven, The Netherlands). After 2 h moths were returned to the laboratory bench in an air-conditioned room at 25 °C, removed from their microcentrifuge tubes, and separately placed on their backs on a white paper surface. The time until recovery, defined as a righting response and regained coordinated muscle function, was recorded (in seconds) [36]. 

#### 2.1.4. Flight Performance

To determine whether larval developmental acclimation impacts adult flight performance in the laboratory, males (*n* = 50) and females (*n* = 50) were exposed to a cold temperature (17 °C) and a benign flight temperature (25 °C) (in a temperature-controlled climate room) for the three acclimation temperatures (15, fluctuating 15–25 and 25 °C) using two flight performance metrics (cylinder and insectary, described in more detail below). This was replicated three times to yield a total sample size of 150 males and 150 females per flight temperature, acclimation temperature and flight performance metric. Moths from different acclimation treatments were marked with powdered fluorescent dye (blue, green and pink) (colours randomised for each replicate) (Day Glo^®^, Cleveland, OH, USA) and were differentiated under a UV light (200–400 nm) (Raytech L8-98CB). Male and female moths were tested separately to ensure that the presence of the opposite sex did not influence the other’s flight performance tests. 

For the flight cylinder assay, which are often used as a quality control measure in SIT programmes [37,38,39], moths were placed in a petri dish (with a diameter of 12 centimetres) inside vertical cylinders (15 cm height, 14 cm diameter) (made from black laminated cardboard) in a dark climate-controlled room (set to 17 or 25 °C). A UV light (Eurolux FL-T8/20W/BL) with a white cardboard backing was mounted 1.2 m (flight distance) above the base of the cylinders [40] to act as an attractant. Moths were allowed to habituate to the petri dish for 15 min before the UV light lure was switched on. After 8 h, the moths were counted and categorized as fliers (the moths that flew towards the lure and were on or near the white board backing) and non-fliers (the moths that stayed behind in the petri dish in the cylinder).

The insectary flight metric was used to determine the flight ability of FCM over a flight obstacle (an inflatable swimming pool) and across a larger but thermally regulated area (17 or 25 °C in the dark). It consisted of an inflatable swimming pool (107 cm diameter) half-filled with water placed in the corner of a temperature-controlled climate room (3.10 m length × 2.51 m width × 2.38 m height). A raised plastic platform (35 cm length × 27 cm width × 6 cm height) was placed in the centre of the inflatable pool, and the moths were placed in Petri dishes (with a diameter of 12 cm) on top of this platform in the dark (habituated for 15 min) and were then allowed to disperse for 12–14 h towards a UV light lure mounted in the opposite corner of the room (mounted 2 m from the ground) (2.93 m linear flight distance). The UV light lure were surrounded by 10 sticky pads (Chempac) arranged in a rectangle with 4 (4 columns × 1 row) above and below the light, and 1 each (1 column × 1 rows) on either side of the light. The number of FCM adults left in the petri-dishes after the 12–14 h trial was counted (representing non-fliers) and the number of FCM that reached the UV light lure and were on or near it (fliers) was counted. For this the moths were released in the evening and counted in the morning to represent the nocturnal flight activity of the FCM. 

Additionally, to determine whether there was a possible link between chill coma recovery and flight performance in FCM, 50 male and 50 female FCM adults from each acclimation regime (15, 15–25 and 25 °C) were exposed to chill coma (0 °C for two hours) and subsequently returned to 25 °C and monitored for recovery. The first 15 to recover (fast responders) and the last 15 to recover (slow responders) were separated, marked with different colours of powdered fluorescent dye, and placed inside 12 cm diameter petri dishes to undergo two flight performance trials (flight cylinders and insectary flight, as described above). This was replicated twice (for a total of 100 males and 100 females per acclimation regime) and the two flight performance metrics (cylinder and insectary) were measured at two flight temperatures (17 °C and 25 °C).

##### Morphometrics

To determine if morphological traits varied between the fliers and non-fliers, and the different thermal treatments in the flight trials, body size and wing size were measured on 5 individuals from each group (i.e., *n* = 5 per flight status (fliers or non-fliers), flight temperature, flight assay method, acclimation temperature and sex). 

Body size was determined after drying moths for 72 h in an oven set to 60 °C. The dry mass was then measured on a microbalance to 0.1 mg accuracy (Mettler Toledo, MS104S/01). Wing size was measured by removing the right forewing as close to the thorax as possible using tweezers and a scalpel, washing the wing in a 20% ethanol solution, after which the wing was submerged in a 10% household bleach solution and a soft bristle paint brush used to gently brush away scales [41]. Finally, the wing was washed again in 20% ethanol, allowed to dry and mounted on a microscope slide using clear nail polish. Photos and measurements of the wings were taken using a Leica DFC 290 fixed digital camera attached to a Leica MZ16A auto montage microscope (Leica, Wetzlar, Germany) and the LEICA Application Suite software version 4.12.0 was used to measure the length and the width of the wing. The width was measured as the distance across the discal cell between the point where the CuA2 vein meets the discal cell and the point where the R1 vein meets the discal cell, whilst the length was measured as the distance across the discal cell between the point where the R1 vein meets the discal cell and the point where the M1 vein meets the discal cell (see Appendix; Appendix A). Due to the pronounced wing fraying after flight, these points were the only consistent discernible points on the wings for measurement. 

The wing and body measurements (body size, wing length and wing width) obtained were then used to estimate wing area, wing loading and aspect ratio using the following equations [42]:

Wing area:S=R×W
where *S* is the wing area of the discal cell (mm^2^), *R* is the wing length (mm) and *W* is the wing width (mm).

Wing loading:$$WL=\frac{Mb}{S}$$
where *WL* is the wing loading (mg mm^−2^), *Mb* is the body size (mg) and *S* is the discal cell wing area (mm^2^).

Aspect ratio:$$AR=\frac{4R^{2}}{S}$$
where *AR* is the aspect ratio, *R* is the wing length (mm), and *S* is the wing area (mm^2^) of the discal cell.

### 2.2. Adult Transport Conditions and Performance

#### 2.2.1. Simulating Transport Conditions

In an effort to simulate transport conditions and their impact on performance, adults were divided into groups of 60 (30 males and 30 females) per pre-treatment, which included a vibration treatment (vibration on a wire rack) and a control (no vibration on a wire rack) at two different temperatures (4 °C (inside a walk-in fridge) and 25 °C (in a climate-controlled room)). Vibrations were simulated using an electrical drill (Bosch GSR 12-2 Professional) as it came the closest to mimicking the vibration experienced during transport; however, the frequency of the vibration for the drill was, on average, higher (ranging between 30 and 50 Hz) than that experienced during transport (ranging between 3 and 5 Hz) and the amplitude of movement during vibration (vertical movement) was lower with the drill compared to the truck (N. Boersma pers. comm.). Since we wanted to investigate whether added vibration influences chill coma recovery time (CCRT) and flight performance these levels were adequate. Moths were placed (*n* = 30 at a time) inside a petri dish (with a diameter of 12 centimetres) inside a plastic container (35 cm length × 27 cm width × 6 cm height) on a wire rack (88 cm width × 46 cm height) and a drill attached to the wire rack next to the plastic container using a rope. Vibrations were transferred through the wire rack and the vibration experienced by the moth was measured using an accelerometer placed on the surface of the plastic container (MetaMotionC, MbientLab) and downloaded using the metabase application (version 3.4.16). The moths were exposed to these treatments for 4 h (the assumed average transport time for the moths shipped from the XSIT rearing facility to the release area (short distance transportation), [N.Boersma pers. Comm.] and were then used within one hour of the treatments to determine chill coma recovery and flight performance. Due to slow emergence rates, all trials were run on 2–4-day old as this is the typical age range used in SIT release programmes [22].

#### 2.2.2. Chill Coma Recovery Time (CCRT) and Spontaneous Behaviour

As we were interested in dissecting the potential drivers of typical variables simulating transport conditions on recovery and subsequent flight performance, CCRT was determined for the 4 treatments (4 °C vibration, 4 °C control, 25 °C vibration, 25 °C control) and for both sexes (males and females). Thirty males and 30 females were tested per treatment by placing individual moths in 1.5 mL microcentrifuge tubes in a plastic bag inside an ice slurry (0 °C) for 2 h. After the 2 h the moths were placed on their backs on a white surface (in a room at 25 °C) in tiny circles (2 cm in radius) and timed for recovery as described earlier (see description in Section 2.1.3. Low-temperature performance). They were given one hour to wake up from chill coma and subsequently scored for spontaneous behaviour (muscle coordination) by monitoring whether the moth moved out of the circle (1) or not (0). This was done to determine the moth’s ability to perform coordinated movement after exposure to chill coma.

#### 2.2.3. Flight Performance

Flight performance trials were carried out with both males (*n* = 30) and females (*n* = 30) after exposure to transport conditions (4 °C vibration, 4 °C control, 25 °C vibration, 25 °C control) using both flight performance assays (cylinder and insectary) previously described. This was replicated 3 times for each experiment to yield a total sample size of 90 males and 90 females per treatment group.

### 2.3. Statistical Analysis

All statistical analyses were performed in R software v. 3.5.1 (R Development Core Team, 2018). Significance levels were set at *p* < 0.05. All Generalized linear models (GLM’s) were performed using the “MASS” [43] package. All GLM results were subsequently analysed with an anova.glm (test = F) for GLM’s with a gaussian distribution and anova.glm (test = Chisq) for binomial and poisson distributions using the “car” package [44]. Post-hoc tests were performed emmeans (estimated marginal means) using the “emmeans” package [45]. All models were simplified using the stepAIC function in the “MASS” package [43]. Both full and minimum adequate models are reported, where applicable. Model assumption for overdispersion was tested using the dispersion test function in the “AER” package [46].

#### 2.3.1. Acclimation Treatments, Life History and Performance

To test the effect of variation in larval acclimation conditions on subsequent adult low temperature and flight performance of FCM, generalized linear models (GLM’s) with a gaussian distribution and an identity link function were used for all traits except the CCRT data. The CCRT data was found to be over-dispersed and a quassi-Poisson GLM with a log link function was used. For low temperature performance, CT_MIN_ and CCRT, respectively, were used as the dependent variable, whilst acclimation group (15, 15–25 or 25 °C), sex (male or female) and their interactions were used as independent variables. For overall flight performance, percentage fliers were used as the dependent variable and flight temperature (17 or 25 °C), acclimation group (15, 15–25 or 25 °C), sex (male or female) and their interactions were used as the independent variables per flight test (cylinder or insectary). Similarly, for flight performance after having undergone chill coma as adults, percentage fliers were the dependent variable and flight test temperature (17 or 25 °C), acclimation group (15, 15–25 or 25 °C), sex (male or female), response time (fast or slow), and their interactions was used as independent variables per flight test (cylinder or insectary). 

The effect of larval acclimation on the life history traits of the resultant adult moths were analysed with GLMs. Egg-laying capacity was analysed using a GLM with a gaussian distribution and an identity link function with the number of eggs laid as the dependent variable and acclimation (15 °C, 15–25 °C or 25 °C) as the independent variable. Adult survival was analysed using a GLM with a binomial distribution and a logit link function with emergence (0 or 1) as the dependent variable and acclimation (15 °C, 15–25 °C or 25 °C) as the independent variable. 

##### Morphometrics

A Pearson’s correlation matrix was drawn between the different morphological parameters (body size, wing length, wing width, wing area, WL and AR) using the “Hmisc” [47] and “corrplot” [48] packages in R, and body size, wing length, wing width and wing area were all found to be positively correlated (see Appendix A); therefore, it was decided to only use body size, WL and AR to determine the impact of sex and acclimation temperature on morphometrics. The impact of larval acclimation temperature and sex on each of the adult body and wing size metrics were then determined using a GLM with a gaussian distribution and an identity link function in which the different metrics (adult body size, WL and AR) were used as the dependent variable and acclimation temperature, sex, and their interactions were used as independent variables. GLM results were subsequently analysed using an anova.glm (test = F).

#### 2.3.2. Flight Performance in Response to Adult Transport Conditions

Chill coma recovery time data in response to vibration and low temperature pre-treatments were over-dispersed and so a GLM with quasi-Poisson distribution and a log link function were used. Treatment, sex and its interaction were used as independent variables and CCRT (time in s) was used as the dependent variable. To determine whether there was a significant difference in spontaneous behaviour between the treatments, a GLM with a binomial distribution and logit link function was used, with treatment, sex and its interactions as independent variables and spontaneous behaviour as the dependent variable. Lastly, flight performance in response to the pre-treatments was tested using a GLM with a gaussian distribution and an identity link function with treatment, sex and its interactions as independent variables and percentage dispersal as the dependent variable.

## 3. Results

### 3.1. Larval Acclimation Treatment Effects on Adult Performance

#### 3.1.1. Life History

The female moths emerging from larvae that were subjected to 15 °C (mean = 50.63, SE = 7.37) and 15–25 °C FTR (mean = 57.87, SE = 9.60) acclimations had higher daily egg-laying capacity compared to the 25 °C group (mean = 19.2, SE = 3.65) (F = 7.93, *p* < 0.0007); however, there was no significant difference in adult emergence between the three acclimation treatments (χ^2^ = 2.48, *p* = 0.29).

#### 3.1.2. Low-Temperature Performance

Larval thermal acclimation treatments had a significant impact on adult low-temperature performance. For CT_MIN_, there was a significant difference between the acclimation treatments, with the 15 °C and fluctuating (15–25 °C) regimes larval acclimations enhancing low-temperature performance when compared with the 25 °C larval acclimation group for both sexes (F = 13.54, *p* < 0.0001) (Figure 2a). Similarly, with respect to the CCRT, there was a significant difference between the three larval acclimation treatments with the adults emerging from the larvae exposed to the 15 °C acclimation showing a significantly faster recovery time compared with the fluctuating and 25 °C groups in the adult females, whilst there were no significant differences in CCRT between the acclimation treatments in the males (Figure 2b).

#### 3.1.3. Flight Performance

For the cylinder flight method there were significantly more fliers at the 25 °C flight temperature as compared with the 17 °C flight temperature (F = 39.88, *p* < 0.0001). Overall, there were no significant differences in dispersal of adults emerging from larvae exposed to the different acclimation treatments (F = 0.26, *p* = 0.78) or between sexes (F = 1.95, *p* = 0.17) (Figure 3a). However, estimated marginal means indicated that larval acclimation had a significant impact on adult flight performance in females, with the 15 °C acclimation group having significantly lower dispersal (fliers) compared with the fluctuating and warm (25 °C) groups at the warm (25 °C) flight temperature (Figure 3a).

For the insectary flight method, adult dispersal (number of fliers counted at the lure) was significantly lower for the cold (17 °C) flight temperature compared to the warm (25 °C) flight temperature (F = 37.71, *p* < 0.0001). Additionally, larval acclimation had a significant impact on adult flight performance, with the 15 °C acclimation group having significantly lower dispersal (fliers) as compared with the fluctuating and warm (25 °C) groups at the warm (25 °C) flight temperature (F = 6.38, *p* < 0.006) (Figure 3). There was also a significant interaction between flight temperature and acclimation temperature (F = 3.58, *p* < 0.05) with no difference in dispersal (fliers) of adults between the acclimation treatments at the cold flight temperature (Figure 3a).

For the FCM adults exposed to chill coma before flight performance tests, there was no significant difference in subsequent flight performance between the slow responders (those that took the longest to wake up from CCRT) and the fast responders (those that woke up the fastest from CCRT) for either the cylinder (F = 0.00, *p* = 1.00) or insectary (F = 0.00, *p* = 1.00) flight methods. Additionally, larval acclimation had no impact on adult flight performance during either the cylinder (F = 1.09, *p* = 0.34) or insectary flight tests (F = 1.77, *p* = 0.19) (Figure 3b). There was, however, a significant difference in dispersal (fliers) between the sexes in the insectary flight method (F = 9.71, *p* < 0.005) with the female fliers dispersing significantly less compared with the males; however, this was not the case in the cylinder flight method (F = 0.00, *p* = 1.00) (Figure 3). Additionally, the moths dispersed significantly less at the cool temperature (17 °C) compared with the warm temperature (25 °C) in the insectary flight method (F = 57.54, *p* < 0.0001); however, this was not the case in the cylinder flight method (F = 3.09, *p* = 0.09) (Figure 3b).

##### Morphometrics

Larval developmental acclimation temperature had a significant impact on adult body mass and WL for females (F = 6.53, *p* < 0.002; F = 1.44, *p* = 0.24) but not for males (F = 2.18, *p* = 0.12; F = 4.34, *p* < 0.02), with the females emerging from larvae exposed to the fluctuating thermal regime (15–25 °C) having smaller bodies and smaller WL compared to those emerging from larvae exposed to the 15 °C acclimation (Figure 4a,b). Females also had a significantly larger body size and WL and AR values compared to males (Figure 4b,c). Furthermore, larger moths had greater WL and AR values (Figure 4b,c).

### 3.2. Adult Transport Conditions and Performance 

#### 3.2.1. Chill Coma Recovery Time (CCRT) and Spontaneous Behaviour

There was a significant difference in CCRT between the treatments (χ^2^ = 16.67, *p* < 0.002) and a significant interaction between treatment and sex with the two sexes responding differently to the treatments (χ^2^ = 35.64, *p* < 0.0001) (Table 1). Females recovered faster at the control conditions (25 °C and no vibration) as compared with the transport conditions (4 °C and vibration); however, this was not the case for males (Figure 5a). Additionally, females also recovered faster at the 25 °C vibration conditions compared with the 4 °C vibration conditions, but once again, this was not the case for males, who recovered significantly faster at the 4 °C vibration conditions compared to all other treatments (Figure 5a).

There was no significant difference in spontaneous behaviour between the treatments (χ^2^ = 2.45, *p* = 0.48) or sexes (χ^2^ = 0.24, *p* = 0.62) (Table 1). 

#### 3.2.2. Flight Performance

No difference was found in flight performance between the adults exposed to the different treatments for the cylinder flight test (F = 0.42, *p* = 0.74). However, there was a significant difference between the sexes (F = 7.65, *p* < 0.02), with males having significantly more fliers as compared with females (Figure 5b). For the insectary flight test there was a significant difference between both the treatments and the sexes (Table 1). Once again, the males had significantly more fliers as compared with females (Figure 5b). Additionally, the females that underwent the 4 °C vibration treatment had significantly more fliers as compared with the other treatments. For males, 4 °C vibration had significantly more fliers compared to 25 °C no vibration and 4 °C no vibration, whilst 25 °C no vibration had more fliers compared to 25 °C vibration (Figure 5b). In contrast, the males in the cylinder flight test had significantly more fliers compared with the insectary tests, and males had significantly more fliers as compared with females (Figure 5b).

## 4. Discussion

Temperature affects a wide array of traits in insects that include life history, morphology, stress tolerance and physical performance [49,50,51]. Adjusting rearing temperature during insect mass-rearing in a species-specific way could lead to offsetting potentially low performance of adults both in the field (increased mating frequency and flight performance) and in the laboratory (increased fecundity and shorter life cycles). This could be a cost-effective and relatively straightforward way to increase the efficacy of an SIT programme [9,29,51]. Here, we determined the impact of rearing temperature (particularly a fluctuating regime) and transport conditions on low temperature and flight performance of adult FCM. Exposure to an FTR during larval development led to both enhanced cold tolerance (CT_MIN_) and increased flight performance of both adult males and females emerging from these larvae whilst still maintaining high egg-laying capacity. In addition, FTR did not have a meaningful impact on pupal development time with preliminary observations indicating only a day difference from first pre-pupa forming to first adult emergence between the control and the FTR (13 vs. 14 days) group, whilst there was a five-day difference observed between the control and the cold treatment (13 vs. 18 days). Boardman et al. [35] showed that exposure of FCM larvae to FTR protected the insects from temperature-induced damage at the cellular and whole-body levels. This was thus expected in this study, as FTR usually gives the benefits of acclimation without incurring the cost of prolonged exposure to an unfavourable temperature [5] and explains why there was no apparent trade-off in physiological performance as a result of FTR during larval development. In addition, the physiological recovery hypothesis states that periodic warm periods during acclimation allows insects to repair damage done during cold acclimation [52,53].

Further beneficial effects of FTR have been found in several other studies [54,55,56,57], with the latter showing that applying FTR to the SIT programme of *Drosophila suzukii* (Matsumura) (Diptera: Drosophilidae) led to decreased senescence and maintained high fecundity for longer. This led to increased *D. suzukii* production. The current study enhanced cold tolerance in moths emerging from larvae acclimated at a low temperature (15 °C) during development, but flight performance was significantly lower compared to adults emerging from larvae exposed to the other acclimation treatments. This indicates a possible trade-off with changes in larval thermal history leading to both costs and benefits to adult performance in the field [51]. Differences in performance as a result of larval acclimation could not be fully explained by morphology. Although significant effects of developmental acclimation could be found in several morphological traits, no consistent trend was visible between acclimation treatments. Similar results were found in *Ceratitis capitata* (Wiedemann) (Diptera: Tephritidae) [13]. In contrast, Frazier et al. [12] showed that an increased flight ability at low temperatures was associated with larger wings as a result of cold acclimation in *D. melanogaster*. However, both Frazier et al. [12] and Shirai [13] reared the insects at a low temperature for the entirety of their developmental period, whilst we restricted acclimation to 5 days of the development period, therefore it could be of use for future investigations to determine the impact of changes in rearing temperature (for the entire development time) on morphology in FCM adults.

To disentangle whether vibration (motorized vehicle) and low-temperature exposure during adult transport in the SIT programme could negatively affect performance of the released FCM in the field, we found a significant difference in CCRT between the four treatments (4 °C vibration, 4 °C control, 25 °C vibration and 25 °C control) with the females showing significantly worse cold tolerance under transport conditions compared to control conditions; however, no trend was discernible in the males. Additionally, we found no significant difference between the treatments for either spontaneous behaviour or flight performance. As a result, the vibration and low temperature experienced by FCM during transport for short distances in the SIT programme did not seem to have a large impact on the performance of adult FCM in the field. This is in contrast to what was found by Nepgen et al. [24], where flight ability was negatively affected by transport when irradiated moths were transported for long distances. However, here the consignments experienced severe changes in temperature during transportation (possibly due to the inability to keep the animals cool for such long periods of time), which possibly led to the decrease in flight ability observed. Therefore, the mode and procedures used by XSIT during transport seems to be adequate to allow for the delivery of high-quality adult moths in the release site; however, temperature regulation to keep the animals cool during transit is very important, particularly during long-distance transport. Neither cold nor fluctuating larval developmental acclimation significantly improved adult flight at a cold ambient temperature (17 °C) and, therefore, it seems that although cold performance and flight performance (at 25 °C flight temperature) were enhanced by developmental acclimation (FTR), FCM remains sensitive to cold ambient temperatures during flight. However, the effect of cold ambient temperatures in the field could possibly be negated through the use of refugia (microclimates) by FCM (see [58] for more information on microclimates). CCRT followed by dispersal did not show any difference in flight performance between the fast and slow responders indicating no link between higher cold tolerance and increased flight performance. Additionally, there was no difference in flight performance between the acclimation treatments, possibly indicating that exposure to sub-lethal temperatures in adults may undo some of the benefits of developmental acclimation [59,60]. If this is the case, low-temperature exposure during transport, may undo the beneficial acclimation response to FTR in the field and it is thus important for future studies to determine flight ability of FCM after exposure to a larval FTR and transport.

In lepidopteran SIT programmes, the release of both males and females are unavoidable due to the inability to separate the insects by gender en masse, as lepidopterans lack an effective genetic sexing strain [61]. Females are generally unwanted in the SIT programme, as they could mate with the released males (reducing mating occurrences between partially sterile males and wild females); they are often damaging and female presence may require a higher radiation dose to assure that they are sterile prior to release, which may be deleterious to male fitness [10,62]. However, several studies indicate that the released sterile females may have a positive impact on population suppression in lepidoptera [10,63]. Therefore, the general increase in both low-temperature and flight performance of the female FCM in our study should further increase the efficacy of the SIT programme by allowing increased mating occurrences between sterile females and wild males in the field. 

It should also be considered that our study only looked at rearing temperature effects for mass reared FCM and did not consider irradiated moths, therefore it would be beneficial to study the impact of FTR on irradiated moths. Additionally, the larvae in our study were only acclimated for 5 days; thus, it would be important to look at these effects after applying different durations of acclimation. Additionally, the impact of larval acclimation on developmental time in our study was only observational and thus it would be useful to quantitatively determine these effects. In addition, this study was laboratory based and replicating the flight performance studies using mark-release-recapture methods could be of use to confirm results in the field, as a weaker effect in response to a temperature pre-treatment for FCM has been found previously [64]. Also, the cylinder flight method (often used as a quality control method in SIT programmes [35,36,37] seems to be a less sensitive metric for indicating differences in flight ability as compared with the insectary method employed in this study. Thus, it may be of use to integrate field and laboratory quality control measures when evaluating the success of a SIT programme [35,36,37].

## 5. Conclusions

Exposure to a fluctuating thermal regime during larval development, even just for a limited time, leads to enhanced performance with minimal costs or trade-offs in FCM. These findings support the implementation of FTR in the SIT programme for FCM as a means of increasing the quality of insects and subsequently improving the efficacy of the programme. Additionally, our study found that SIT transport conditions did not have a large impact on low-temperature or flight performance of FCM and, thus, we encourage its continued use, granted temperature is tightly controlled during transport.

## Figures and Tables

**Figure 1 insects-13-00315-f001:**
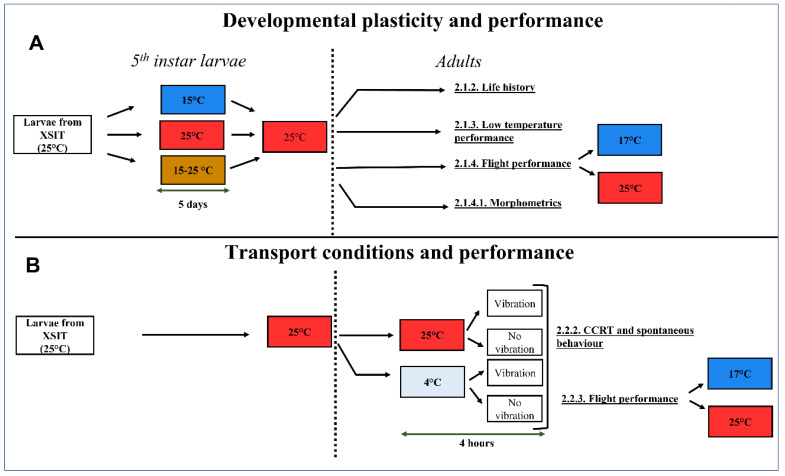
Schematic diagram of experimental protocols showing 5th instar FCM larvae experiencing (**A**) developmental acclimation (representing a change in rearing conditions) or (**B**) transport conditions (adults experiencing temperature and vibration pre-treatments). Several traits were measured to assess the effects of these treatments on performance of FCM. Details of the experiments are presented in Section 2.1 and Section 2.2. (XSIT- X Sterile Insect Technique (Pty) Ltd, Citrusdal, South Africa.; CCRT—chill coma recovery time).

**Figure 2 insects-13-00315-f002:**
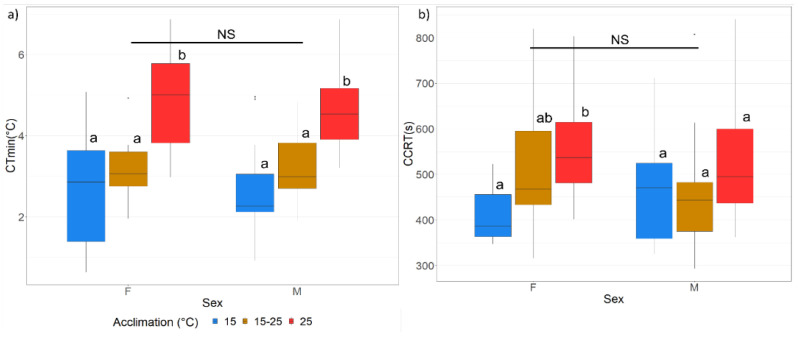
(**a**) Boxplot indicating median CT_MIN_ (°C), upper and lower quantiles and maximum and minimum values (whiskers) for adult males (M) and females (F) from larvae exposed to the three acclimation temperatures (15 °C, 15–25 °C and 25 °C); (**b**) Boxplot indicating median CCRT (s), upper and lower quantiles and maximum and minimum values (whiskers) for adult males (M) and females (F) emerging from larvae exposed to the three acclimation temperatures (15, 15–25 and 25 °C). Significant differences between sexes are indicated as: * = *p* < 0.05, ** = *p* < 0.001, *** = *p* < 0.0001, NS = not significant. Different letters indicate significant differences between acclimation groups per sex.

**Figure 3 insects-13-00315-f003:**
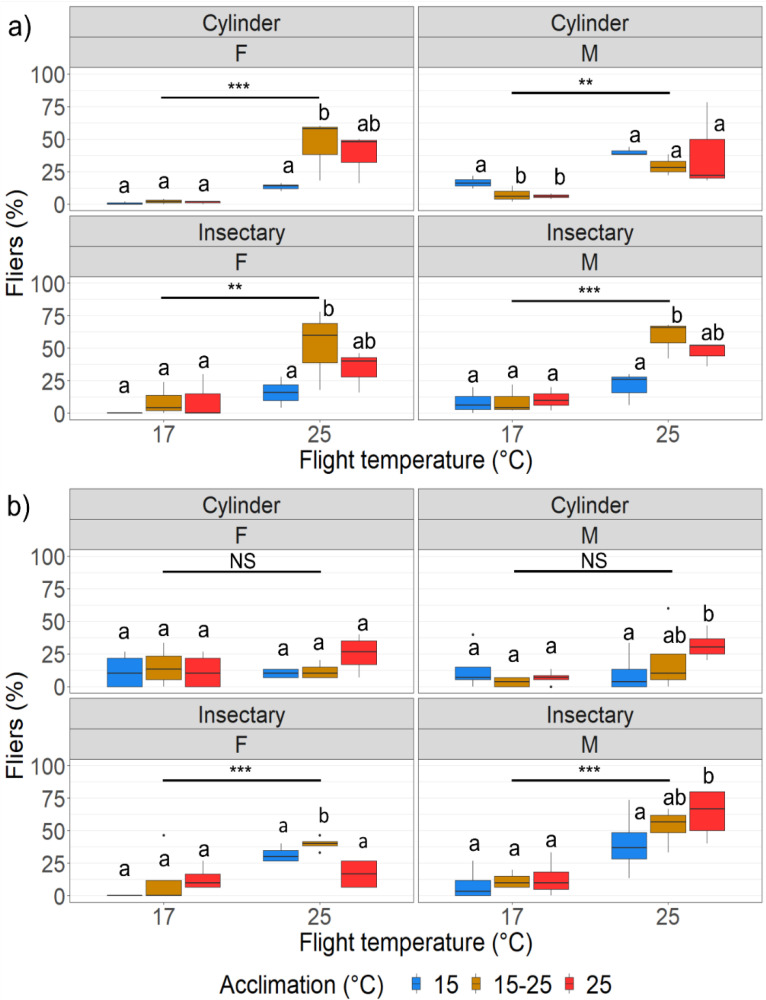
(**a**) Boxplot indicating median adult male (M) and female (F) fliers (%), upper and lower quantiles and maximum and minimum values (whiskers) for the two flight temperatures (17 and 25 °C), and adults from the larvae from the three larval acclimation temperatures (15, 15–25 and 25 °C) for the cylinder and insectary flight tests; (**b**) boxplot indicating subsequent adult male (M) and female (F) median fliers (%), upper and lower quantiles and maximum and minimum values (whiskers) after their exposure to chill coma for the two flight temperatures (17 and 25 °C) for adults from the larvae from the three larval acclimation temperatures (15, 15–25 and 25 °C) for the cylinder and insectary flight tests; significant differences between flight temperatures are indicated as: * = *p* < 0.05, ** = *p* < 0.001, *** = *p* < 0.0001, NS = not significant. Different letters indicate significant differences between acclimation temperatures per flight temperature.

**Figure 4 insects-13-00315-f004:**
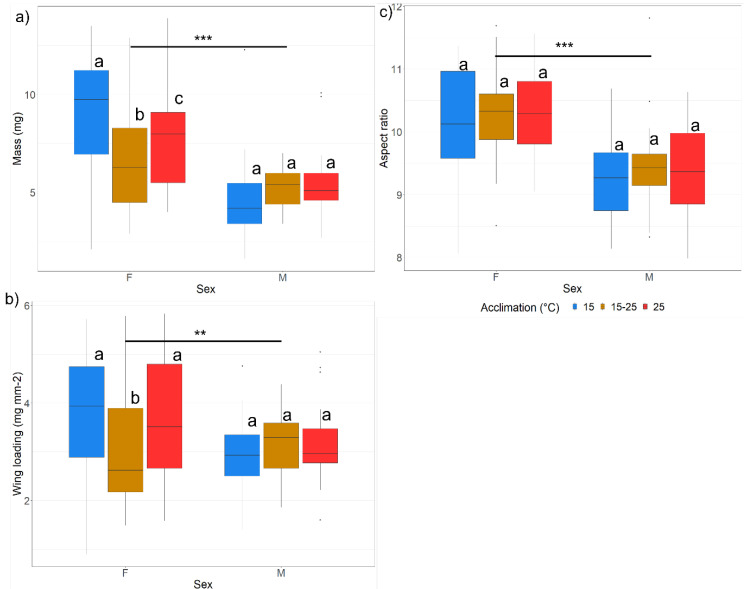
Boxplots indicating (**a**) mass (mg); (**b**) wing loading (mg mm^−2^); (**c**) aspect ratio with their upper and lower quantiles and maximum and minimum values (whiskers) for adult males (M) and females (F) emerging at the three different acclimation temperatures (15°C, 15–25°C and 25 °C). Significant differences between flight temperatures are indicated as: * = *p* < 0.05, ** = *p* < 0.001, *** = *p* < 0.0001, NS = not significant. Different letters indicate significant differences between acclimation per flight temperature.

**Figure 5 insects-13-00315-f005:**
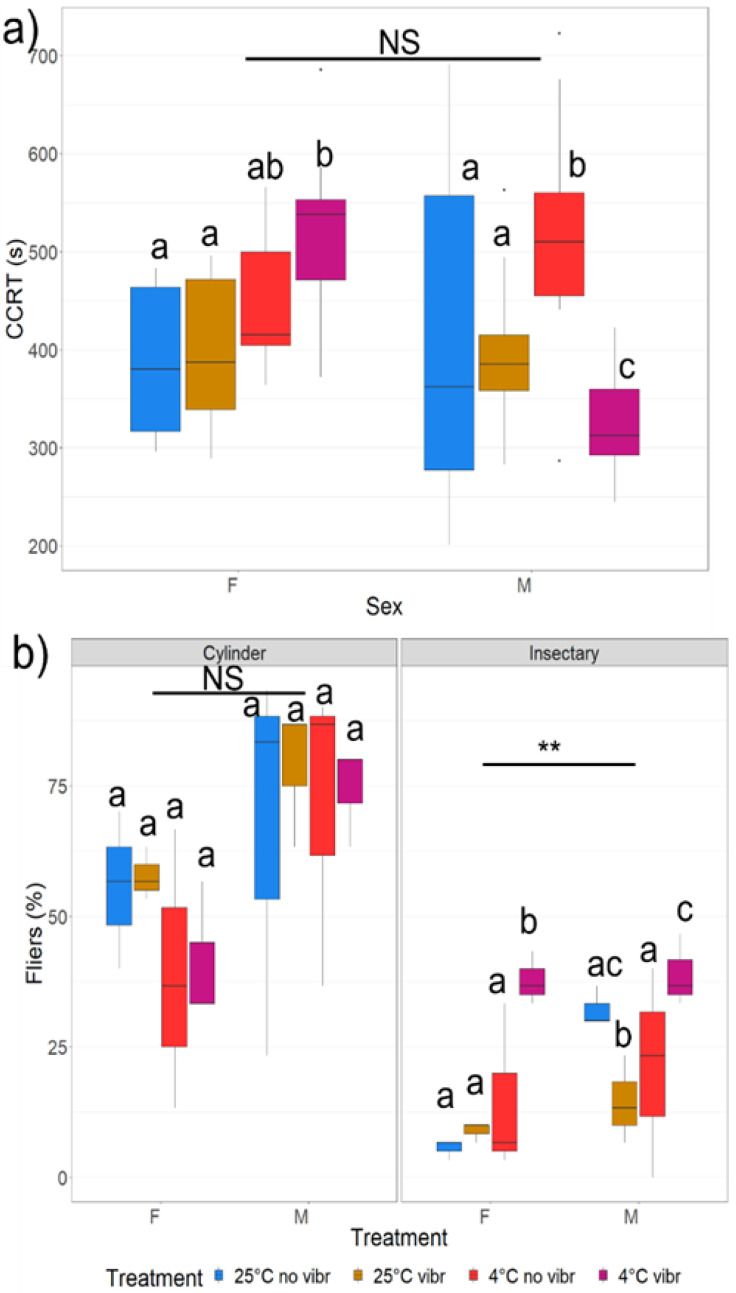
Boxplots indicating (**a**) median CCRT (s) of adult males (M) and females (F) with their upper and lower quantiles and maximum and minimum values (whiskers) after exposure to vibration treatments (25 °C no vibration, 25 °C vibration, 4 °C no vibration and 4 °C vibration); and (**b**) flight performance (median adult male (M) and female (F) fliers (%)) with their upper and lower quantiles and maximum and minimum values (whiskers) at the two different flight tests (cylinder and insectary) after exposure to vibration treatments (25 °C no vibration, 25 °C vibration, 4 °C no vibration and 4 °C vibration). Significant differences between sexes are indicated as: * = *p* < 0.05, ** = *p* < 0.001, *** = *p* < 0.0001, NS = Not significant. Different letters indicate significant differences between treatments.

**Table 1 insects-13-00315-t001:** Results of the full (saturated) model of the generalized linear model investigating chill coma recovery time, spontaneous behaviour and flight performance following vibration and temperature pre-treatments showing the F-value (flight performance) or Chi-square value (χ^2^) (spontaneous behaviour), degrees of freedom, residual degrees of freedom and the *p*-value.

Chill Coma Recovery Time (CCRT)	χ^2^	df	*p*-Value	
Treatment	16.6740	3	<0.002	
Sex	0.8560	1	0.3549	
Treatment × Sex	35.6400	3	<0.0001	
Spontaneous behaviour	χ^2^	df	*p*-value	
Treatment	2.4510	3	0.4842	
Sex	0.2414	1	0.6232	
Treatment × Sex	1.6119	3	0.6567	
Flight performance: Cylinder	F	df	Resid df	*p*-value
Treatment	0.4242	3	20	0.7383
Sex	7.6522	1	19	<0.02
Treatment × Sex	0.3505	3	16	0.7894
Flight performance: Insectary	F	df	Resid df	*p*-value
Treatment	7.62	3	20	<0.003
Sex	5.7093	1	19	<0.03
Treatment × Sex	1.8443	3	16	0.1798

## Data Availability

The data presented in this study are available upon reasonable request from the corresponding author.

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
