# Peer review of "Consequences of Thermal Variation during Development and Transport on Flight and Low-Temperature Performance in False Codling Moth (Thaumatotibia leucotreta): Fine-Tuning Protocols for Improved Field Performance in a Sterile Insect Programme"

_insects, 2022, doi:10.3390/insects13040315_

Round 1

Reviewer 1 Report

This manuscript describes a series of well designed and appropriately analyzed experiments that address important practical questions relating regarding effects of exposure to low temperature during rearing and to low temperature and vibration during shipping of false codling moths to be released in a sterile insect technique program. The aim is to provide information necessary to optimizing rearing and handling protocols for use in the ongoing SIT program for false codling moth in South Africa. The context for the study, the methodology, results, conclusions, and relevance are all clear described. The conclusions are well supported by the data.  Overall, although narrow in scope, the findings are important and have relevance beyond the false codling moth system and the optimization of protocols used in this specific SIT program. 

I have only a few minor comments to offer:

Figure 1: It would be helpful to the reader if acronyms used in the figure are spelled out in the figure caption (e.g., XSIT, CCRT)

Line 355: “Egg laying capacity were analyzed . . .”  It appears the analysis described is specifically for adult emergence and the analysis of egg laying capacity is missing.  Egg laying capacity is measured as number of eggs produced per female and cannot be analyzed as a binomial outcome.  Survival of each individual to adult (=emergence – yes or no) is measured as a binomial outcome and appropriately analyzed as described.

Line 388: Provide summary results (means and SE) for the egg laying capacity for each treatment in the text. These data are not presented anywhere.

Author Response

Comment 1: This manuscript describes a series of well designed and appropriately analyzed experiments that address important practical questions relating regarding effects of exposure to low temperature during rearing and to low temperature and vibration during shipping of false codling moths to be released in a sterile insect technique program. The aim is to provide information necessary to optimizing rearing and handling protocols for use in the ongoing SIT program for false codling moth in South Africa. The context for the study, the methodology, results, conclusions, and relevance are all clear described. The conclusions are well supported by the data.  Overall, although narrow in scope, the findings are important and have relevance beyond the false codling moth system and the optimization of protocols used in this specific SIT program.

Response: Thank you.

Comment 2: Figure 1: It would be helpful to the reader if acronyms used in the figure are spelled out in the figure caption (e.g., XSIT, CCRT).

Response: We have changed the figure caption to spell out any acronyms used in the figure.

Comment 3: Line 355: “Egg laying capacity were analyzed . . .”  It appears the analysis described is specifically for adult emergence and the analysis of egg laying capacity is missing.  Egg laying capacity is measured as number of eggs produced per female and cannot be analyzed as a binomial outcome.  Survival of each individual to adult (=emergence – yes or no) is measured as a binomial outcome and appropriately analyzed as described.

Response: This was a typo - apologies. Egg laying capacity was in fact analysed using a GLM with a gaussian distribution whilst emergence was analysed using a GLM with a binomial distribution (0 and 1). We have rectified this mistake in the text (Line 354).

Comment 4: Line 388: Provide summary results (means and SE) for the egg laying capacity for each treatment in the text. These data are not presented anywhere.

Response: Summary results for egg laying capacity for each treatment was added to the text (Line 391).

Reviewer 2 Report

The manuscript addresses issues with the physical factors (low temp. handling) during transport and rearing of false codling moth for SIT. The primary finding was that fluctuating temps during larval development enhanced performance.

The ms was amazingly free of typos. The statistical analysis was appropriate. The experimental design was well conceived. The conclusions drew logically from the results. A very professional job. 

Author Response

Comment: The manuscript addresses issues with the physical factors (low temp. handling) during transport and rearing of false codling moth for SIT. The primary finding was that fluctuating temps during larval development enhanced performance.

The ms was amazingly free of typos. The statistical analysis was appropriate. The experimental design was well conceived. The conclusions drew logically from the results. A very professional job.

Response: Thank you for the positive comments.